# Periprosthetic fractures: the next fragility fracture epidemic? A national observational study

Alex Bottle [1], Richard Griffiths,[2] Stuart White,[3] Henry Wynn-Jones,[4] Paul Aylin,[1] Iain Moppett [5] Emyr Chowdhury,[6] Helen Wilson,[7] Benjamin M Davies [8]

► Prepublication history and supplemental material for this paper is available online. To view these files, please visit the journal online (http://dx.doi.org/10.1136/bmjopen-2020-042371).

For numbered affiliations see end of article.

**Correspondence to**
Professor Alex Bottle;
robert.bottle@imperial.ac.uk

## ABSTRACT

**Objectives** Periprosthetic fractures have considerable clinical implications for patients and financial implications for healthcare systems. This study aims to determine the burden of periprosthetic fractures of the lower and upper limbs in England and identify any factors associated with differences in treatment and outcome.

**Design** A national, observational study.

**Setting** England.

**Participants** All individuals admitted to hospital with periprosthetic fractures between 1 April 2015 and 31 December 2018.

**Primary and secondary outcome measures** Mortality, length of stay, change in rate of admissions.

**Methods** We analysed Hospital Episode Statistics data using the International Classification of Diseases 10th Revision code M96.6 (Fracture of bone following insertion of orthopaedic implant, joint prosthesis, or bone plate) to identify periprosthetic fractures recorded between April 2013 and December 2018. We determined the demographics, procedures performed, mortality rates and discharge destinations. Patient characteristics associated with having a procedure during the index admission were estimated using logistic regression. The annual rate of increase in admissions was estimated using Poisson regression.

**Results** Between 1 April 2015 and 31 December 2018, there were 13565 patients who had 18888 admissions (89.5% emergency) with M96.6 in the primary diagnosis field. There was a 13% year-on-year increase in admissions for periprosthetic fracture in England during that period. Older people, people living in deprived areas and those with heart failure or neurological disorders were less likely to receive an operation. 14.4% of patients did not return home after hospital discharge. The overall inpatient mortality was 4.3% and total 30-day mortality was 3.3%.

**Conclusions** The clinical and operational burden of periprosthetic fractures is considerable and increasing rapidly. We suggest that the management of people with periprosthetic fractures should be undertaken and funded in a similar manner to that successfully employed for people sustaining hip fractures, using national standards and data collection to monitor and improve performance.

## Strengths and limitations of this study

► This is the first study to use International Classification of Diseases 10th Revision code M96.6 to identify trends in periprosthetic fracture rates.
► These data can be used to help develop specific management pathways and allocate resources.
► Use of Hospital Episode Statistics data is limited by potential inaccuracies in coding.
► The study data do not include functional outcomes, which would provide useful information.

## INTRODUCTION

Since the successful introduction of total hip replacement in the 1960s, joint replacement has revolutionised the treatment of severely arthritic joints.[1] Joint replacement prostheses are available for all the major joints of the upper and lower limbs. In 2018, approximately 200000 arthroplasty procedures were recorded in the 16th Annual Report of the National Joint Registry (NJR) for England and Wales.[2] Between the advent of the registry, on 1 April 2003, and 31 December 2018, 2.75 million joint replacement procedures have been recorded. In addition, 28684 femoral hemiarthroplasties were performed in 2018 for proximal femoral fractures.[3] The increasing life expectancy in the UK,[4] combined with the effects of chronic disease and frailty,[5] leads to an increased likelihood of sustaining a fracture due to osteoporosis and an increased incidence of falls. Fractures can also occur around joint replacements, and these are termed periprosthetic fractures. Periprosthetic fractures may require either fixation or revision of the implant. Surgery to treat such fractures is often complex and/or prolonged, requiring specialist input from a multidisciplinary team.

There has been no comprehensive assessment of the burden of these injuries on the healthcare system in England. The NJR collects data on revision arthroplasty and identifies those performed for periprosthetic fracture. However, fractures treated without the need to revise the implant are not recorded at present in the NJR. The alternative sources

of data are the UK's four national hospital administrative databases: Hospital Episode Statistics (HES) for England and similar systems for each of the other three countries. Clinical coders use International Classification of Diseases 10th Revision (ICD-10) and UK-specific Office of Population Censuses and Surveys (OPCS) coding systems to record diagnosis and procedure information, respectively. Owing to the complex classification of fractures and different techniques to treat them, there are many operative codes that could be used. In March 2015, National Health Service (NHS) England issued new guidance to clinical coders regarding periprosthetic fractures.[6] The new ICD-10 code M96.6 ('Fracture of bone following insertion of orthopaedic implant, joint prosthesis, or bone plate') is designed to capture all patients who have experienced a periprosthetic fracture at any time during an episode of care, regardless of site or operative treatment. It also captures fractures around fixation devices such as plates and intramedullary nails.

The purpose of this observational study is to quantify the burden of these cases on the NHS and identify any trends in the treatment of these conditions, which may inform future healthcare resource planning. Using England's national hospital administrative database, HES, we describe the admissions with the new M96.6 diagnosis code, time trends, patient characteristics, operative rate and short-term outcomes. A literature review carried out by the authors did not identify any previous reporting of these national-level data.

## METHODS
### Data
HES[7] comprises over 125 million admitted patient, outpatient and emergency department records from hospitals within England's NHS annually. Inpatient records consist of consultant episodes: each episode covers the period during which a patient is the responsibility of a given physician. A single admission at a given hospital may comprise multiple episodes. Episodes were linked into admissions and linkage with interhospital transfers creates 'superspells'; we will use the term 'admission' throughout to refer to superspells. HES provides data on in-hospital mortality and is linked to the national death register. One primary and up to 19 secondary diagnoses are coded using ICD-10[8]; one primary and up to 23 secondary procedures are coded using the UK's own 'OPCS' coding system. Area-level socioeconomic deprivation status in population-weighted quintiles was linked to HES via the patient's postcode. Coding is completed by specialist clinical coders (rather than clinicians) in hospitals from the medical record. Once received centrally the data are cleaned to remove duplicates.

We extracted all admissions from HES with the ICD-10 code M96.6 in the primary diagnosis field (which corresponds to the main problem treated) and discharge dates between April 2013 and December 2018.

## Outcome measures
We report the primary procedure performed, length of stay (LOS) (acute hospital and superspell), final destination on discharge, in-hospital mortality rate and total 30-day mortality rate.

We also report the proportion of admissions with a procedure, excluding scans via the U chapter of OPCS (Diagnostic imaging, testing and rehabilitation), urinary catheterisation and auxiliary or minor procedures via the X, Y and Z chapters of OPCS (Miscellaneous and subsidiary classification of methods and sites of operation).

## Statistical analysis
With any coding change, some variation in the uptake rate between hospitals is expected. We tried to separate the uptake of the new code from any genuine underlying change in the number of admissions by dividing hospital trusts into 'early adopters' and the rest. 'Early adopters' were defined as those with two or more discharges with M96.6 as the primary diagnosis in March or April 2015. Simple Poisson regression models estimated the linear trend in admissions for each group of trusts ('early adopters' and 'other') and for all combined. As a sensitivity analysis the modelling was repeated after excluding the first 6 months of the new coding period, that is, including only admissions from October 2015 onwards.

Patient characteristics were described for the cohort from April 2015 including: age, gender, method of admission (whether emergency or not), area-level socioeconomic deprivation[9] and comorbidity (using the HES ICD-10 secondary diagnosis codes). The proportion of admissions with a procedure coded, as defined above, was noted and the factors associated with a procedure derived from a multivariable logistic regression model with age, gender, deprivation and comorbidities as covariates entered at once with no selection or elimination. To determine what factors were associated with an extended LOS a multivariable logistic regression was performed with the same predictors as for having a procedure; again, all predictors were retained in the model irrespective of their p value. We performed the usual diagnostics for logistic regression, including checking for collinearity between predictors, goodness of fit and residuals. These were satisfactory. In the absence of an accepted definition of extended LOS in these cases we used the upper quartile boundary (26) to define a long LOS. Kaplan-Meier curves were plotted for overall survival since admission.

## Patient and public involvement
This research was done without patient involvement. As an observational study of national-level data it was not possible to involve these groups. The suggested next steps of designing services to treat these injuries appropriately would benefit from patient and public involvement.

**Table 1** Total numbers of admissions with M96.6 recorded since April 2013

| Financial year (1 April to 31 March) | Numbers of admissions with M96.6 as primary diagnosis | Numbers of admissions with M96.6 only as secondary diagnosis |
| --- | --- | --- |
| 2013/2014 | 429 | 384 |
| 2014/2015 | 514 | 415 |
| 2015/2016 | 3832 | 974 |
| 2016/2017 | 5232 | 1077 |
| 2017/2018 | 5435 | 1123 |
| 2018/2019 to 31 December 2018 | 4389 (5852 full-year pro rata) | 893 (1191 full-year pro rata) |
| Total | 19 831 | 4866 |

## RESULTS

Between 1 April 2013 and 31 December 2018, there were 24 697 admissions with M96.6 recorded in any diagnosis field. After the coding guidance was introduced in March 2015, there were 22 955 such admissions (84.5% classified as emergency) for 16 105 patients. For the 80.3% who had M96.6 in the primary diagnosis field, there were 18 888 admissions (89.5% emergency) for 13 565 patients.

### Change in admissions over time

Table 1 gives the number of admissions by year, split by whether M96.6 was in the primary or in the secondary diagnosis fields.

The subsequent analysis is restricted to admissions with M96.6 in the primary diagnosis.

Figure 1 illustrates admissions by month since April 2014 to December 2018, with elective and emergency

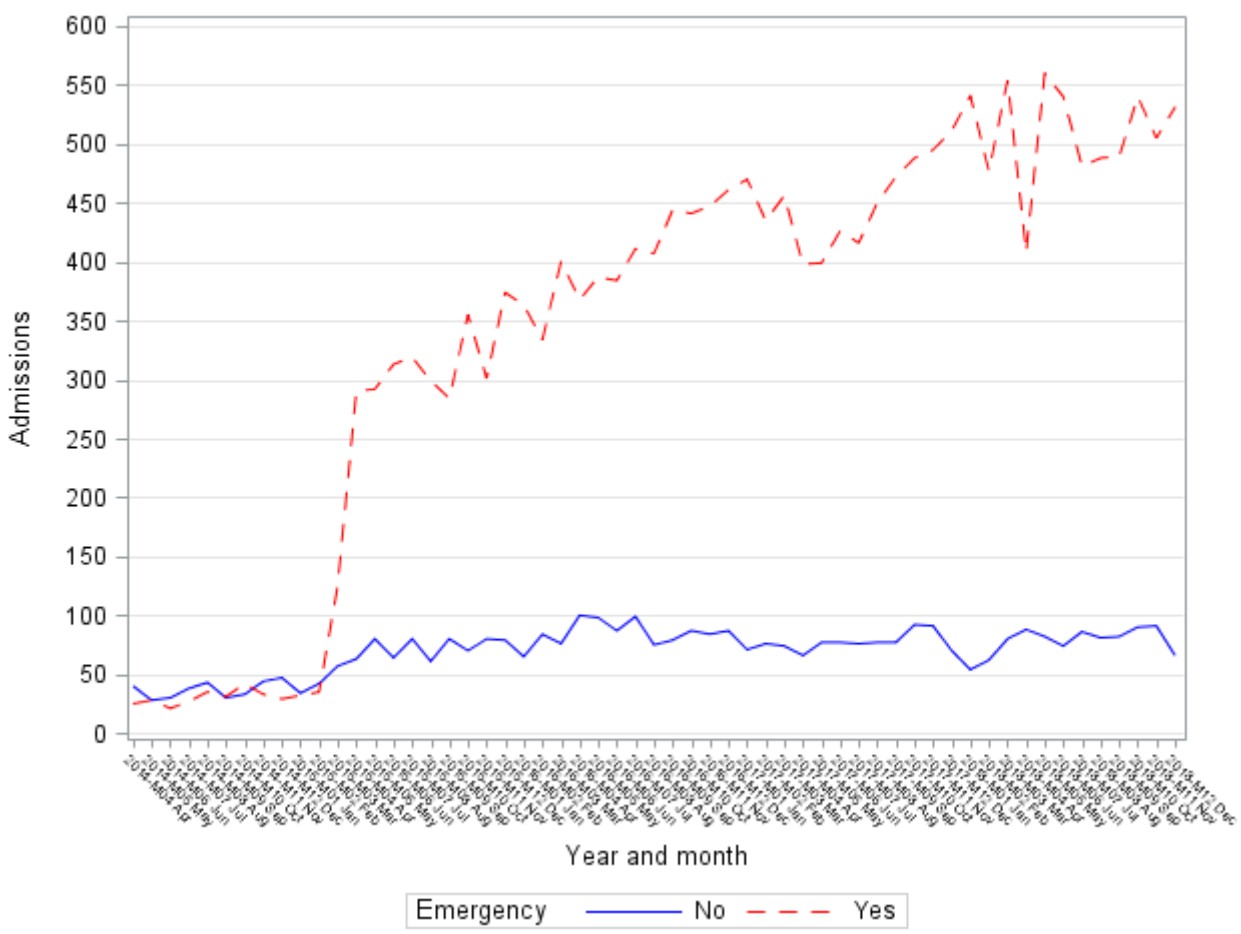

**Figure 1** Number of admissions by month from April 2013 to December 2018, with elective and emergency admissions shown separately.

**Table 2** Poisson regression showing annual increase in use of M96.6 diagnosis code nationally and split by whether hospitals were early adopters of this code

| Set of trusts | Relative risk per month | 95% CI | Relative risk per year | 95% CI |
|---|---|---|---|---|
| All | 1.011 | 1.010 to 1.012 | 1.14 | 1.13 to 1.15 |
| Early M96.6 adopters | 1.010 | 1.009 to 1.011 | 1.13 | 1.11 to 1.14 |
| Non-early adopters | 1.016 | 1.013 to 1.018 | 1.21 | 1.17 to 1.24 |
| All (October 2015 onwards only) | 1.009 | 1.008 to 1.010 | 1.11 | 1.10 to 1.13 |

admissions shown separately. The rapid rise in admissions from April 2015 coincides with the introduction of the M96.6 coding guidance.

One hundred sixteen out of 141 hospital trusts using the M96.6 code were labelled early adopters. The rate of increase (slope) for the non-early adopters was significantly higher than that for the early adopters (p=0.001 for the interaction term). Taking the early adopters as being the 'true' underlying trend, there was a 13% annual rise in M96.6 admissions between April 2015 and December 2018 (table 2). Excluding the first 6 months since the coding change did not change the rate of increase significantly.

### Patient characteristics
The majority of admissions were for women (66.4% female, 33.6% male). Table 3 details patient characteristics.

### Operative rate and main procedures performed
Given that the coding advice only came into force in March 2015, we now restrict the analysis to the 11 019 discharges from April 2015 onwards, again with M96.6 as the primary diagnosis.

Primary open reduction (W201+W192+W202+W191+W198) was the most common recorded surgical technique (21.6%); revision of the prosthesis (W373+W383+W384+W374+W403+W393+W404) was performed in 10.4% (the most common procedures are listed in online supplemental table 1).

The overall operative rates were 74.7% in 2015/2016, 72.1% in 2016/2017 and 74.5% in 2017/2018. For the elective admissions, 83.5% had a procedure performed; for the emergency admissions, 71.8% had a procedure performed, with an overall proportion of 73.0%.

Table 4 gives the operative rate by patient characteristic and the logistic regression results. (See online supplemental table 2 for operative rates by age, gender and method of admission.) Patients of both sexes aged greater than 84 years were significantly less likely to have an operation than the youngest age group. Surgery was also less likely with increasing socioeconomic deprivation, chronic obstructive pulmonary disease (COPD) and neurological disorders. Surgery was more likely in the presence of recorded fluid and electrolyte disturbance, hypertension and obesity (OR 1.78, 95% CI 1.46 to 2.17).

### Length of stay
Median acute hospital LOS was 14 nights (IQR 7–26), and median total hospital LOS was 17 nights (IQR 8–33: online supplemental table 3). Acute hospital LOS increased with increasing age and was higher for older ages for both sexes and for operative than non-operative management at each age group (p<0.001; figures 2 and 3 and online supplemental tables 3–5).

In multivariable logistic regression, increasing age was significantly associated with a longer LOS. A wide variety of comorbidities were also associated with a longer LOS, but with a smaller effect than that of age (see table 5).

### Final destination on discharge
89.3% of identified patients were admitted from their usual residence and 74.9% went back there afterwards: 8.9% went to a care home; 5.3% were discharged to a temporary residence; 4.3% died in hospital; and 3.6% went to another hospital but for which no HES record could be linked.

The total 30-day mortality rate was 3.3% for the period 1 April 2015 to 30 June 2018 (deaths via linkage to the national death registry were available up to dates including 31 July 2018).

Figure 4 shows Kaplan-Meier survival curves for non-operative versus operative cases. See online supplemental figures 1 and 2 for Kaplan-Meier survival curves by sex and age range, respectively.

### DISCUSSION
This study sought to establish the burden of periprosthetic fractures on the healthcare system in the UK and identify any factors associated with differences in treatment or outcome. The number of patients admitted to hospital in England with periprosthetic fractures is currently around 450 per month. The overall number of periprosthetic fractures as coded by M96.6 is increasing by approximately 13% each year. Periprosthetic fractures are associated with a relatively prolonged LOS and increased dependency on discharge, with 25% unable to return directly to their normal residence. Seventy-three per cent of people admitted with a periprosthetic fracture received an operation. Non-operative management was more common in those with comorbidities, increasing age and COPD. Mortality was low at 4.3% for in-hospital

**Table 3** Patient characteristics

| Factor | Admissions | Proportion of total admissions (%) |
|---|---|---|
| Age, sex | | |
| 0–44, female | 215 | 1.1 |
| 0–44, male | 468 | 2.5 |
| 45–64, female | 1068 | 5.7 |
| 45–64, male | 792 | 4.2 |
| 65–84, female | 6186 | 32.8 |
| 65–84, male | 3424 | 18.1 |
| 85+, female | 5063 | 26.8 |
| 85+, male | 1672 | 8.9 |
| Deprivation | | |
| Quintile 1 (least deprived) | 4796 | 25.4 |
| Quintile 2 | 4467 | 23.6 |
| Quintile 3 | 4090 | 21.7 |
| Quintile 4 | 3143 | 16.6 |
| Quintile 5 (most deprived) | 2300 | 12.2 |
| Quintile 6 (not known) | 92 | 0.5 |
| Dementia | 2634 | 13.9 |
| Alcohol misuse | 625 | 3.3 |
| Arrhythmias | 4130 | 21.9 |
| Heart failure | 1445 | 7.7 |
| COPD | 3337 | 17.7 |
| Deficiency anaemia | 493 | 2.6 |
| Depression | 1231 | 6.5 |
| Diabetes mellitus | 2877 | 15.2 |
| Fluid disorders | 1396 | 7.4 |
| Hypertension | 9208 | 48.8 |
| Hypothyroidism | 1804 | 9.6 |
| Liver disease | 229 | 1.2 |
| Cancer with metastasis | 251 | 1.3 |
| Obesity | 784 | 4.2 |
| Other neurological conditions | 1204 | 6.4 |
| Paraplegia | 180 | 1.0 |
| Pulmonary circulatory disorders | 204 | 1.1 |
| Peripheral vascular disease | 520 | 2.8 |
| Renal disease | 2189 | 11.6 |
| Rheumatological disorders | 1619 | 8.6 |
| Solid tumours | 650 | 3.4 |
| Valvular disease | 1091 | 5.8 |

COPD, chronic obstructive pulmonary disease.

and 3.3% for total 30-day mortality; for proximal femoral fractures, 30-day mortality from the latest National Hip Fracture Database (NHFD) report is higher at 6.1%.[3]

There are limitations to the data, and we have tried not to overinterpret the data we have. We cannot be certain how complete and accurate the M96.6 coding is. The study relies on accurate data recorded by clinicians and then coded by clinical coders in hospitals. The primary diagnosis and procedure fields in administrative data are known to have high accuracy (>95%),[10] though secondary diagnoses are subject to some under-recording. There will be variation between hospitals as to when the coding advice was implemented, which is why we estimated the trend in two ways. We have attempted to correct for this by analysing 'early adopters' separately from the complete cohort. Reassuringly, the conclusions are broadly the same in terms of absolute numbers and relative rate of rise. We did not attempt to distinguish between the sites of periprosthetic fractures, and some of these fractures will have been following arthroplasty of other joints, and fracture fixation, particularly neck of femur fracture.[11] M96.6 is not associated with an anatomical code to specify the bone and/or joint involved. It is possible to attempt to infer the joint involved by any subsequent operation codes, but this method is not guaranteed to be accurate and does not provide any information regarding the site of injury of those who were treated non-operatively (approximately 27% of all cases). Nonetheless, despite these limitations, we are confident that the increasing trend among older people with joint implants accurately reflects the everyday clinical situation.

HES data only provide in-hospital mortality, but we were able to use the established linkage to the national death registry for deaths up to the end of July 2018. More importantly, there are no direct data on functional outcomes.[12] Such additional data would be invaluable to allow clinicians and patients to discuss the most appropriate management.

This is the first study using the specific clinical code M96.6. This was introduced in March 2015, and although figure 1 shows a dramatic increase when the coding advice was introduced, we are confident that there is a significant increasing underlying trend in patients with periprosthetic fractures. There have been few studies that have examined the periprosthetic patient population from a multidisciplinary perspective. There are no studies examining risk assessment for these patients with the exception of a small retrospective analysis, 39 patients with distal femoral fractures from Cleveland, Ohio.[13]

Our data are in line with previous studies of complication rates, which have demonstrated periprosthetic fracture rates of 0.1%–18% for hip, 0.3%–5.5% for knee and 0.5%–3% for shoulder arthroplasties.[14]

The number of periprosthetic fractures is likely to increase, given the ageing population, increased life expectancy and increasing numbers of people with joint replacements. How these individuals are managed is one of the issues that needs to be addressed in order to provide optimum care. The likely treatment of these injuries varies depending on the location involved and the stability of the implant.

**Table 4** Operative rate by patient characteristics and multivariable logistic regression results

| Factor | No operation | Operation | Proportion with operation (%) | Adjusted OR for operation | P value |
|---|---|---|---|---|---|
| Age 00–44, female | 52 | 163 | 75.8 | 1.23 (0.83 to 1.81) | 0.299 |
| Age 00–44, male | 97 | 371 | 79.3 | 1 | |
| Age 45–64, female | 203 | 865 | 81.0 | 1.29 (0.91 to 1.83) | 0.158 |
| Age 45–64, male | 152 | 640 | 80.8 | 1.27 (0.88 to 1.82) | 0.206 |
| Age 65–84, female | 1576 | 4610 | 74.5 | 0.86 (0.62 to 1.19) | 0.371 |
| Age 65–84, male | 822 | 2602 | 76.0 | 0.97 (0.69 to 1.34) | 0.830 |
| Age 85+, female | 1674 | 3389 | 66.9 | 0.6 (0.44 to 0.83) | 0.002 |
| Age 85+, male | 524 | 1148 | 68.7 | 0.67 (0.48 to 0.94) | 0.019 |
| Quintile 1 (least deprived) | 1267 | 3529 | 73.6 | 1 | |
| Quintile 2 | 1183 | 3284 | 73.5 | 1.00 (0.91 to 1.10) | 0.998 |
| Quintile 3 | 1082 | 3008 | 73.5 | 0.98 (0.89 to 1.08) | 0.742 |
| Quintile 4 | 882 | 2261 | 71.9 | 0.89 (0.80 to 0.98) | 0.025 |
| Quintile 5 (most deprived) | 655 | 1645 | 71.5 | 0.84 (0.75 to 0.95) | 0.004 |
| Quintile 6 (not known) | 31 | 61 | 66.3 | 0.64 (0.41 to 1.01) | 0.053 |
| Alcohol misuse | 151 | 474 | 75.8 | 0.93 (0.76 to 1.14) | 0.484 |
| Arrhythmias | 1208 | 2922 | 70.8 | 0.96 (0.89 to 1.05) | 0.385 |
| COPD | 879 | 2458 | 73.7 | 0.84 (0.74 to 0.96) | 0.008 |
| Cancer with metastasis | 65 | 186 | 74.1 | 1.02 (0.93 to 1.11) | 0.681 |
| Deficiency anaemia | 150 | 343 | 69.6 | 0.89 (0.73 to 1.08) | 0.229 |
| Dementia | 910 | 1724 | 65.5 | 0.88 (0.77 to 1.00) | 0.055 |
| Depression | 319 | 912 | 74.1 | 1.02 (0.89 to 1.16) | 0.830 |
| Diabetes mellitus | 777 | 2100 | 73.0 | 0.98 (0.89 to 1.07) | 0.613 |
| Fluid disorders | 340 | 1056 | 75.6 | 1.31 (1.15 to 1.49) | <0.001 |
| Heart failure | 466 | 979 | 67.8 | 1.13 (1.06 to 1.21) | 0.0004 |
| Hypertension | 2424 | 6784 | 73.7 | 1.07 (0.95 to 1.19) | 0.264 |
| Hypothyroidism | 483 | 1321 | 73.2 | 0.98 (0.71 to 1.35) | 0.907 |
| Liver disease | 56 | 173 | 75.5 | 1.03 (0.75 to 1.42) | 0.868 |
| Obesity | 125 | 659 | 84.1 | 1.78 (1.46 to 2.17) | <0.001 |
| Other neurological conditions | 362 | 842 | 69.9 | 0.80 (0.70 to 0.91) | 0.001 |
| Peripheral vascular disease | 143 | 377 | 72.5 | 0.87 (0.63 to 1.21) | 0.401 |
| Paraplegia | 54 | 126 | 70.0 | 1.28 (0.92 to 1.79) | 0.145 |
| Pulmonary circulatory disorders | 49 | 155 | 76.0 | 1.00 (0.82 to 1.23) | 0.974 |
| Renal disease | 637 | 1552 | 70.9 | 0.98 (0.88 to 1.09) | 0.695 |
| Rheumatological disorders | 455 | 1164 | 71.9 | 0.90 (0.80 to 1.01) | 0.065 |
| Solid tumours | 182 | 468 | 72.0 | 0.96 (0.79 to 1.17) | 0.699 |
| Valvular disease | 297 | 794 | 72.8 | 1.11 (0.96 to 1.28) | 0.161 |

COPD, chronic obstructive pulmonary disease.

The extensive surgery involved in the operative treatment options for these individuals means that these patients have to be able to withstand a more significant surgical insult than individuals undergoing primary joint arthroplasty or those being treated for neck of femur fractures. There are also implications for the use of hospital resources, with longer operation times, expensive implants and the availability of revision arthroplasty specialists in addition to the higher cost of postoperative care, including high dependency care.[15]

In-hospital mortality is significant at around 5%, and the rate of new discharge to a care home of around 15% suggests an increase in dependency of this patient group—this represents those who, while admitted from their usual residence, were discharged to a different location.

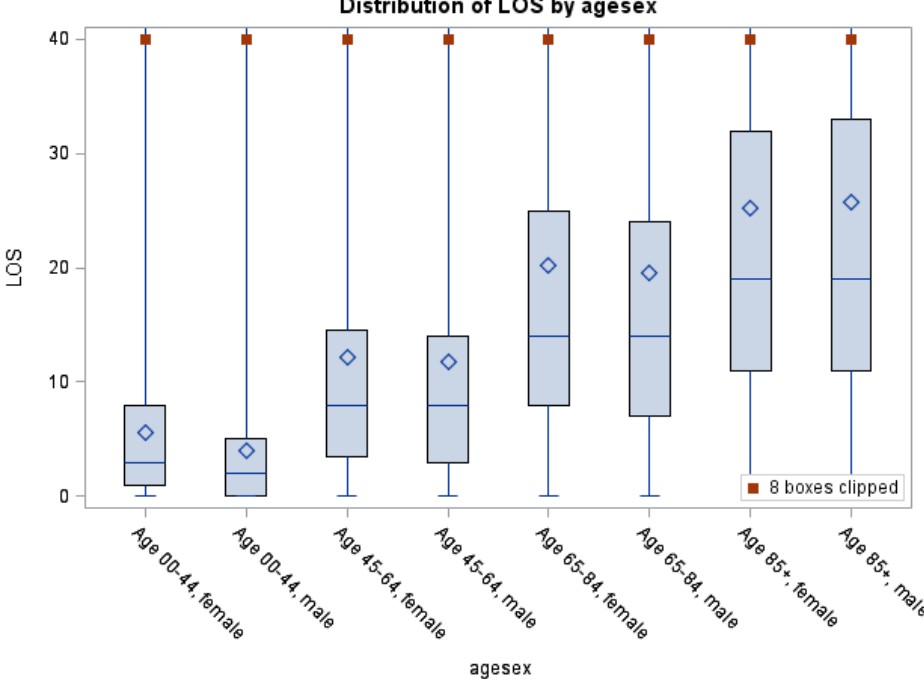

**Figure 2** Acute hospital length of stay (LOS) by age and gender.

For patients presenting with hip fracture, surgical management is almost always viewed as gold standard care for pain relief and to restore mobility where possible. Early surgery is important to allow the patient to sit up, thereby reducing risk of respiratory problems, malnutrition and difficulty in toileting. This may be different in periprosthetic fractures, particularly those around the knee, as patients may be able to tolerate sitting up and pain may be controlled by immobilising the fracture in a cast or brace.

Some patients with frailty do not have the physiological reserve to survive major surgery, and this will influence decisions between non-operative management, fixation and major revision surgery.

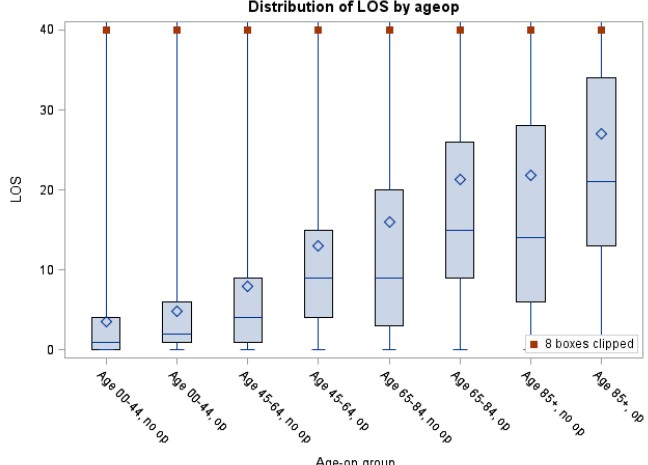

**Figure 3** Acute hospital length of stay (LOS) by operative management.

For all patients with frailty, either cognitive or physical, the decision regarding surgery should be made with the individual and the multidisciplinary team and in some cases may be viewed as a palliative procedure to manage pain.

Notwithstanding the limitations of our analysis, periprosthetic fractures are likely to create an increasing burden to patients, clinicians and the wider health and social care services. At current rates, the average UK trust will admit around one patient with a periprosthetic fracture each week. If the current trend of a 13% annual increase continues, this will rise to around two patients per week in 5 or 6 years' time, equating to around 700 extra bed-days per trust per year, alongside an increase in required operating theatre capacity. These patients are often complex, both surgically and medically, and hospitals may wish to consider whether a specialist local or regional service is provided to ensure optimal decision-making and management. The costs associated with treating periprosthetic fractures often outstrip reimbursement from funders, and so any such service will need to be designed in conjunction with funders to ensure it is financially viable.[16]

Data collection and accurate reporting are going to be needed to identify all of the cases, and an add-on to the NJR is required for complete data capture. If a comparison with proximal femoral fracture is made, the NHS is probably where it was with hip fracture care around 15 years ago as far as the care of periprosthetic fractures is concerned. The great improvement in hip fracture care ran in parallel with the emergence of 'orthogeriatrics' as a definite entity. Periprosthetic fractures will need the

**Table 5** Factors associated with a long LOS (≥26 nights)

| Factor | OR (95% CI) | P value |
|---|---|---|
| Age 00–44, female | 0.66 (0.21 to 2.12) | 0.489 |
| Age 00–44, male | 1 | |
| Age 45–64, female | 4.41 (1.77 to 10.95) | 0.001 |
| Age 45–64, male | 3.97 (1.58 to 9.95) | 0.003 |
| Age 65–84, female | 11.01 (4.52 to 26.86) | <0.001 |
| Age 65–84, male | 9.95 (4.07 to 24.32) | <0.001 |
| Age 85+, female | 17.93 (7.35 to 43.74) | <0.001 |
| Age 85+, male | 17.00 (6.93 to 41.66) | <0.001 |
| Quintile 1 (least deprived) | 1 | |
| Quintile 2 | 1.03 (0.93 to 1.13) | 0.603 |
| Quintile 3 | 1.02 (0.92 to 1.13) | 0.689 |
| Quintile 4 | 1.09 (0.98 to 1.22) | 0.115 |
| Quintile 5 (most deprived) | 1.15 (1.02 to 1.30) | 0.027 |
| Quintile 6 (not known) | 0.88 (0.50 to 1.53) | 0.642 |
| Alcohol misuse | 1.14 (0.92 to 1.42) | 0.221 |
| Arrhythmias | 1.27 (1.17 to 1.38) | <0.001 |
| COPD | 1.32 (1.17 to 1.50) | <0.001 |
| Cancer with metastasis | 1.10 (1.00 to 1.20) | 0.039 |
| Deficiency anaemia | 1.44 (1.18 to 1.74) | 0.003 |
| Dementia | 1.43 (1.26 to 1.63) | <0.001 |
| Depression | 1.10 (0.96 to 1.27) | 0.187 |
| Diabetes mellitus | 1.19 (1.09 to 1.31) | 0.0002 |
| Fluid disorders | 2.29 (2.04 to 2.56) | <0.001 |
| Heart failure | 1.03 (0.96 to 1.10) | 0.493 |
| Hypertension | 1.00 (0.89 to 1.12) | 0.949 |
| Hypothyroidism | 2.08 (1.54 to 2.81) | <0.001 |
| Liver disease | 0.83 (0.60 to 1.15) | 0.270 |
| Obesity | 1.25 (1.05 to 1.48) | 0.012 |
| Other neurological conditions | 1.47 (1.28 to 1.69) | <0.001 |
| Peripheral vascular disease | 2.12 (1.53 to 2.92) | <0.001 |
| Paraplegia | 1.73 (1.29 to 2.32) | 0.0003 |
| Pulmonary circulatory disorders | 1.05 (0.86 to 1.28) | 0.652 |
| Renal disease | 1.20 (1.08 to 1.33) | 0.0004 |
| Rheumatological disorders | 1.17 (1.04 to 1.32) | 0.012 |
| Solid tumours | 1.68 (1.39 to 2.04) | <0.001 |
| Valvular disease | 1.30 (1.13 to 1.49) | 0.0002 |

COPD, chronic obstructive pulmonary disease; LOS, length of stay.

same level of multidisciplinary input that is adequately funded and using all the experience gained with the hip fracture population.

Will individual units be able to cope with demand or will there be regional specialisation? The resources needed will not be limited to orthopaedic surgery, where there will need to be enough adequately trained arthroplasty revision surgeons, but will include perioperative medicine, orthogeriatric input, high dependency care, nursing, physiotherapy and care home provision. Perhaps, dedicated units that deal with hip fractures and periprosthetic fractures will be the model of care for the future, with expertise and training opportunities concentrated in purpose-built units.

The coding change made in 2015 to make the NHS ICD-10 compliant has made the identification of periprosthetic fractures easier. There already appears to be an increase of 13% per year in these cases. This requires

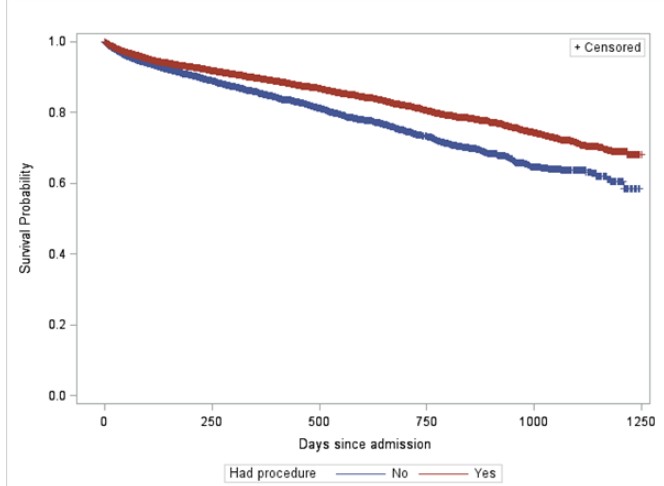

**Figure 4** Kaplan-Meier survival curve by operative status.

planning for future service provision at local and national levels. This may be facilitated by prospective data collection using a system similar to the NJR or NHFD.

**Author affiliations**
[1]Dr Foster Unit, Department of Primary Care and Public Health, School of Public Health, Imperial College London, London, UK
[2]Department of Anaesthesia, Peterborough City Hospital, Peterborough, UK
[3]Haywards Heath Hip Fracture Unit, Princess Royal Hospital, Brighton and Sussex University Hospitals NHS Trust, Haywards Heath, UK
[4]The Centre for Hip Surgery, Wrightington Hospital, Hall Lane, Wigan, UK
[5]Anaesthesia and Critical Care, Division of Clinical Neuroscience, The University of Nottingham, Queen's Medical Centre, Nottingham, UK
[6]Department of Trauma and Orthopaedics, Peterborough City Hospital, Peterborough, UK
[7]Department of General Medicine, Stroke and Care of the Elderly, Royal Surrey County Hospital, Guildford, UK
[8]Division of Trauma and Orthopaedics, Addenbrooke's Hospital, University of Cambridge, Cambridge, UK

**Acknowledgements** We would like to thank Ketan Patel for his assistance regarding the use of the code M96.6.

**Contributors** AB devised the study, extracted and analysed the data, wrote the manuscript and reviewed the manuscript. RG devised the study, reviewed the data, wrote the manuscript and reviewed the manuscript. SW, HWJ, PA, IM and HW reviewed the data, wrote the manuscript and reviewed the manuscript. EC devised the study, reviewed the data and reviewed the manuscript. BMD devised the study, reviewed the data, wrote the manuscript, reviewed the manuscript and guaranteed the paper.

**Funding** This publication presents independent research funded by the National Institute for Health Research (NIHR). AB and PA work at the Dr Foster Unit, an academic unit in the Department of Primary Care and Public Health, Imperial College London. The unit receives research funding from Dr Foster Intelligence, an independent health service research organisation (a wholly owned subsidiary of Telstra). The Dr Foster Unit at Imperial College London is affiliated with the NIHR Imperial Patient Safety Translational Research Centre. The NIHR Imperial Patient Safety Translational Research Centre is a partnership between the Imperial College Healthcare NHS Trust and Imperial College London. The Department of Primary Care and Public Health at Imperial College London is grateful for the support from the NW London NIHR Collaboration for Leadership in Applied Health Research and Care (CLAHRC) and the Imperial NIHR Biomedical Research Centre. BMD is funded by an NIHR Clinical Lecturer position (CL-2016-14-009).

**Disclaimer** The views expressed are those of the author(s) and not necessarily those of the NHS, the NIHR or the Department of Health and Social Care. The funders did not have any input to study conception, design, analysis or manuscript writing.

**Competing interests** AB reports grants from Dr Foster Unit, grants from Medtronic, outside the submitted work. HWJ has received payment in past for educational and consultation services to DePuy Synthes and Stryker; institutional grants received from DePuy Synthes for international surgeon visitations; Trustee of The John Charnley Trust.

**Patient consent for publication** Not required.

**Ethics approval** We have approval from the Secretary of State and the Health Research Authority under Regulation 5 of the Health Service (Control of Patient Information) Regulations 2002 to hold confidential data and analyse them for research purposes (CAG ref 15/CAG/0005). We have approval to use them for research and measuring quality of delivery of healthcare, including for this analysis, from the London-South East Ethics Committee (REC ref 15/LO/0824).

**Provenance and peer review** Not commissioned; externally peer reviewed.

**Data availability statement** Data may be obtained from a third party and are not publicly available. Due to information governance rules applicable to HES, no data are available for sharing. HES data are available directly from NHS Digital. Please see: https://digital.nhs.uk/data-and-information/data-tools-and-services/data-services/hospital-episode-statistics/users-uses-and-access-to-hospital-episode-statistics#access-to-data.

**ORCID iDs**
Alex Bottle http://orcid.org/0000-0001-9978-2011
Iain Moppett http://orcid.org/0000-0003-3750-6067
Benjamin M Davies http://orcid.org/0000-0002-4878-0697

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
