## [Reviewer comments · BMJ Open]

ARTICLE DETAILS

TITLE (PROVISIONAL)	Periprosthetic fractures – the next fragility fracture epidemic? A national observational study
AUTHORS	Bottle, Alex; Griffiths, Richard; White, Stuart; Wynn-Jones, Henry; Aylin, Paul; Moppett, Iain; Chowdhury, Emyr; Wilson, Helen; Davies, Benjamin

VERSION 1 – REVIEW

REVIEWER	Baha John Tadros East Kent Hospitals NHS Trust
REVIEW RETURNED	16-Apr-2020

GENERAL COMMENTS	The paper sets out to report on the burden of periprosthetic fractures on hospitals. This questions has been answered appropriately. However I disagree with the inferences made regarding the trend. I have no doubt that the rate of periprosthetic fractures is on the rise, but this can't be extrapolated from the results presented as the differences in numbers between each year are minor, in addition to the inaccuracy of coding.
---

REVIEWER	David N. Bernstein, MD, MBA, MEI Harvard Combined Orthopaedic Residency Program
REVIEW RETURNED	09-May-2020

GENERAL COMMENTS	Thank you for the opportunity to review an original research study submitted to BMJ Open entitled, "Periprosthetic fractures – the next fragility fracture epidemic? A national observational study". I read the manuscript with great interest. Using a national (UK) database, the authors sought to determine trends in in periprosthetic fractures in England from 2015 to 2018. They conclude that the clinical and operational burden of periprosthetic fractures is large and only increasing over time. Overall, I find this national, observational study quite interesting. However, I have a number of key concerns that substantially reduce my enthusiasm for the manuscript in its current form: 1) I believe the key study questions/goals should be more clearly outlined in the Abstract and at the end of the Introduction. While I appreciate the authors demonstrating a number of key trends/findings, the manuscript in its current form appears to be a fishing expedition - at times - of a large database in search for important findings. The end of the Introduction is vague and very broad. Thus, perhaps it would be better to state the questions
---

more clearly. For example: 1) What is the trend of hospital admissions for periprosthetic fractures from April 2013 to December 2018? Etc. By clearly outlining the questions/goals in this way, readers will have a better sense early on what the main takeaways of the paper will likely be.

2) While I appreciate the authors attempting to quantify the changes appreciated by the coding guidance suggesting the use of M96.6 in 2015, the manuscript, as written, is a bit confusing because of it. For example, part of the Abstract notes trends began to be analyzed in April 2015; however, the Methods section in the Abstract and in the manuscript notes April 2013. I believe only data from April 2015 onward should be included to limit confusion; the limitations can then note that acceptance of this guidance across institutions may have differed and the findings could be underestimated because of it. I do not necessarily believe the additional analyses on this topic add much or are warranted, and if the authors are concerned about such an approach, one could start the analysis from six months following guidance being released.

3) The core findings of this study provide important national epidemiological information, but to be honest, are not surprising and provide little actionable insight. As the population ages and more patients undergo arthroplasty procedures, it makes sense that there would be a growing number of periprosthetic fractures. The study would be substantially strengthened by analyzing what patient factors are associated with a higher LOS, risk of complication, mortality rate, or discharge location. There is a bit of this in the present study. Currently, however, the study appears substantially more descriptive in nature. Can the authors undertake these important additional analyses and make the results clearer?

ADDITIONAL COMMENTS BY SECTION:

#ABSTRACT

1. As noted above, please clarify the exact study questions or goals; as currently written, they are a bit too broad in this reviewer's opinion.

2. Please clarify your statistical tests and subsequent results -- the authors state that obese patients, for example, are 78% more likely to receive an operation. This suggest an odds ratio was calculated using a logistic regression analysis -- however, very little is noted about this in the manuscript and the results are only in an appendix table. As noted in my major comments above, perhaps such analyses with all key outcomes (operative treatment, LOS, mortality, complications, etc.) being independent factors in their own regressions would be of value.

3. Please remove "relatively" in the conclusion -- relative to what? This either needs to be clear or the authors should simply note that it is increasing.

4. The final sentence in the Conclusions is confusing -- can you please re-word/explain?

#ARTICLE SUMMARY

1. I would argue that if more robust analyses are done that determine what are risk factors for important outcomes, than resources can be better aligned to serve high risk patients. As is, this is hard to truly acknowledge based on the trends presented.

2. Agree that lack of functional outcomes/PROMs is a weakness, as well as long-term outcomes and QoL measures.

#INTRODUCTION

1. The authors provide a great deal of good information here. However, can they please try to condense? I believe it would be easier for readers to include one paragraph on the background known; one paragraph on the limitations of current knowledge/gaps in knowledge; and then one paragraph setting up and then stating the exact study questions/hypotheses.

#METHODS

1. Can the authors provide more detail on the HES? Are the data entered by clinicians? Is it an administrative database? Does anyone audit/check it for accuracy?

2. As noted earlier, I would consider simply going from post-guidance release to December 2018 -- including from April 2013, to me, is misleading and confusing. Further, it likely heavily biases all results if you used the data from April 2013 to April 2015.

3. As noted in my core concerns, there is an opportunity for more robust statistical tests and reporting to help determine more actionable information. For example, evaluating factors associated with increased LOS, mortality, complications, etc.

#RESULTS

1. I do not believe the Poisson regression evaluating early and non-early adopters is of notable value to the argument of the manuscript; please consider removing. Ultimately, the authors can note this as a limitation and suggest their findings may even be underrepresentation of the true rates.

2. Please make the logistic regression results a core table in the manuscript; this is important information and simply hand picking a couple of ORs to report introduces bias.

3. If the p value is > 0.05 (as it is with patients with dementia), please simply state there was no difference. Stating there is a "weaker signal" is misleading.

4. The descriptive information LOS, final destination of discharge, etc. is important; however, more robust analyses to determine association between patient characteristics and these outcomes would add notably more value to this work.

5. Much of the tables/figures are reporting of descriptive statistics (e.g., 72% of patients with rheumatological disorders underwent surgery); however, without understanding the relationship via a regression model of this factor with others, this doesn't really add much insight or help align limited resources.

	#DISCUSSION 1. I believe the Discussion would benefit from a first paragraph that restates - in one sentence or two - the background for the study, followed by stating the goal(s) of the study, and then ending with the key findings that answer the study questions. As it currently written, a lot of good information is present in the first paragraph, but the authors already begin to compare their findings to other findings in the literature. 2. Overall, the Discussion has a great deal of important information; however, as is, I find the Discussion to be a bit wordy and not have a logical flow. The authors discuss common areas of periprosthetic fracture, etc. but their study did not look at this specifically. It may be possible to comment on it as a reason for a result they found, but describing this in detail, to this reviewer, does not seem necessary. In reality, I believe the authors would benefit from structuring their Discussion to have one paragraph for each question/study goal stated. This would make it clearer and more concise for readers. 3. Throughout the Discussion, the authors should refer to the literature; I do not think having a separate section named "Comparison with previous studies" is warranted. 4. In addition to some of the limitations noted, I believe there are a few others that are important to consider: 1) the lack of PROMs/functional outcomes/QoL measures limit the complete understanding of the impact of periprosthetic fractures on patients/outcomes; 2) long-term results (outside of mortality linked through the national death registry) are unavailable from my understanding of the database [please correct me if I'm wrong]; 3) I agree that the lack of ability to discern between the location of the periprosthetic fracture is a large limitation; however, hip and knee are most common/likely. Can the authors cite literature noting this to be the case, so it is clear to the readers what the majority of periprosthetic fractures considered in this study are likely associated with? 5. The authors conclude their Discussion with good insight into potential policy implications. As noted earlier, I believe more actionable insight can be gained from this large dataset about who may or may not need greater resources, etc. This insight can help in the planning and care process more so than simply saying there is an increasing number of periprosthetic fractures. 6. Please state areas for future work/research. Overall, I believe this manuscript includes some interesting national (UK) findings related to periprosthetic fractures. However, a number of major revisions/edits, in this reviewer's opinion, are needed before it would be suitable for publication in BMJ Open. Thank you for the opportunity to review your work, and hope you and yours are staying safe during this ongoing global pandemic.
--	--

REVIEWER	Can Doruk Basa Tepecik Training and Research Hospital, Department of Orthopaedics and Traumatology, Izmir, TURKEY
-----------------	--

	Sports traumatology Knee Shoulder Arthroscopy High Tibial Osteotomy
REVIEW RETURNED	18-May-2020

GENERAL COMMENTS	This study haven't designed properly. Because the upper and lower extremity peri-prosthetic fractures are not similar in nature. And also, the code "M96.6" includes fractures following bone plate too. It is not appropriate to evaluate trauma patients and arthroplasty patients in the same study. Nevertheless, the title, discussion and introduction of the study flow through peri-prosthetic fractures. Since these patients are demographic and socioeconomically different, it is not appropriate to determine their characteristics in the same study. The results of the study do not contribute to either the orthopedist or the epidemiologist from this perspective.
---

VERSION 1 – AUTHOR RESPONSE

Reviewer: 1

Reviewer Name

Baha John Tadros

Institution and Country

East Kent Hospitals NHS Trust

Please state any competing interests or state 'None declared':

None declared

Please leave your comments for the authors below

The paper sets out to report on the burden of periprosthetic fractures on hospitals. This questions has been answered appropriately.

However I disagree with the inferences made regarding the trend.

I have no doubt that the rate of periprosthetic fractures is on the rise, but this can't be extrapolated from the results presented as the differences in numbers between each year are minor, in addition to the inaccuracy of coding.

We would, respectfully, disagree with the reviewer that the data does not allow for extrapolation as it demonstrates a clear trend. We would also suggest that although there are well established issues with coded data, the insights that it can offer make its usage valid. We have addressed the issues surrounding coded data within our discussion of the limitations of the study.

Reviewer: 2

Reviewer Name

David N. Bernstein, MD, MBA, MEI

Institution and Country

Harvard Combined Orthopaedic Residency Program

Please state any competing interests or state 'None declared':

None Declared.

Please leave your comments for the authors below

Thank you for the opportunity to review an original research study submitted to BMJ Open entitled, "Periprosthetic fractures – the next fragility fracture epidemic? A national observational study". I read the manuscript with great interest.

Using a national (UK) database, the authors sought to determine trends in in periprosthetic fractures in England from 2015 to 2018. They conclude that the clinical and operational burden of periprosthetic fractures is large and only increasing over time.

Overall, I find this national, observational study quite interesting. However, I have a number of key concerns that substantially reduce my enthusiasm for the manuscript in its current form:

1) I believe the key study questions/goals should be more clearly outlined in the Abstract and at the end of the Introduction. While I appreciate the authors demonstrating a number of key trends/findings, the manuscript in its current form appears to be a fishing expedition - at times - of a large database in search for important findings. The end of the Introduction is vague and very broad. Thus, perhaps it would be better to state the questions more clearly. For example: 1) What is the trend of hospital admissions for periprosthetic fractures from April 2013 to December 2018? Etc. By clearly outlining the questions/goals in this way, readers will have a better sense early on what the main takeaways of the paper will likely be.

This has been addressed in line with the specific comments made below. Please see the additional comments section for Page/Line references.

2) While I appreciate the authors attempting to quantify the changes appreciated by the coding guidance suggesting the use of M96.6 in 2015, the manuscript, as written, is a bit confusing because of it. For example, part of the Abstract notes trends began to be analyzed in April 2015; however, the Methods section in the Abstract and in the manuscript notes April 2013. I believe only data from April 2015 onward should be included to limit confusion; the limitations can then note that acceptance of this guidance across institutions may have differed and the findings could be underestimates because of it. I do not necessarily believe the additional analyses on this topic add much or are warranted, and if the authors are concerned about such an approach, one could start the analysis from six months following guidance being released.

The inclusion of data prior to April 2015 is important to help identify the both the effect of the adoption of the code M96.6 and to help in establishing the difference between early and late adopters.

3) The core findings of this study provide important national epidemiological information, but to be honest, are not surprising and provide little actionable insight. As the population ages and more patients undergo arthroplasty procedures, it makes sense that there would be a growing number of periprosthetic fractures. The study would be substantially strengthened by analyzing what patient factors are associated with a higher LOS, risk of complication, mortality rate, or discharge location. There is a bit of this in the present study. Currently, however, the study appears substantially more descriptive in nature. Can the authors undertake these important additional analyses and make the

results clearer?

Additional information regarding length of stay (LOS) has been added in line with the specific comments below. The identification of the increase rate of peri-prosthetic fractures provides important information to drive changes in practice as argued in the Discussion.

ADDITIONAL COMMENTS BY SECTION:

#ABSTRACT

1. As noted above, please clarify the exact study questions or goals; as currently written, they are a bit too broad in this reviewer's opinion.

Edited. Line 56 and 143

2. Please clarify your statistical tests and subsequent results -- the authors state that obese patients, for example, are 78% more likely to receive an operation. This suggests an odds ratio was calculated using a logistic regression analysis -- however, very little is noted about this in the manuscript and the results are only in an appendix table. As noted in my major comments above, perhaps such analyses with all key outcomes (operative treatment, LOS, mortality, complications, etc.) being independent factors in their own regressions would be of value.

Edited. Obesity reference removed from abstract. Details of tests used are contained in the methods (rather than the 300 word limited abstract).

3. Please remove "relatively" in the conclusion -- relative to what? This either needs to be clear or the authors should simply note that it is increasing.

Edited. Line 84

4. The final sentence in the Conclusions is confusing -- can you please re-word/explain?

Edited. Line 86

#ARTICLE SUMMARY

1. I would argue that if more robust analyses are done that determine what are risk factors for important outcomes, then resources can be better aligned to serve high risk patients. As is, this is hard to truly acknowledge based on the trends presented.

Done. Details of higher risk patients included in main results/discussion

2. Agree that lack of functional outcomes/PROMs is a weakness, as well as long-term outcomes

and QoL measures.

#INTRODUCTION

1. The authors provide a great deal of good information here. However, can they please try to condense? I believe it would be easier for readers to include one paragraph on the background known; one paragraph on the limitations of current knowledge/gaps in knowledge; and then one paragraph setting up and then stating the exact study questions/hypotheses.

Edited. Three sections removed from the introduction. Questions made clearer in lines 142-147

#METHODS

1. Can the authors provide more detail on the HES? Are the data entered by clinicians? Is it an administrative database? Does anyone audit/check it for accuracy?

Short explanation added regarding the origin of HES data. Lines 194-196

2. As noted earlier, I would consider simply going from post-guidance release to December 2018 -- including from April 2013, to me, is misleading and confusing. Further, it likely heavily biases all results if you used the data from April 2013 to April 2015.

The inclusion of earlier data is partially designed to highlight the effect of the M96.6 code, which is one of the aims of the paper.

3. As noted in my core concerns, there is an opportunity for more robust statistical tests and reporting to help determine more actionable information. For example, evaluating factors associated with increased LOS, mortality, complications, etc.

LOS regression model added. Starts at Line 323, including Table 5

#RESULTS

1. I do not believe the Poisson regression evaluating early and non-early adopters is of notable value to the argument of the manuscript; please consider removing. Ultimately, the authors can note this as a limitation and suggest their findings may even be underrepresentation of the true rates.

We believe this highlights some of the issues around coding which is one of the aims of the paper

2. Please make the logistic regression results a core table in the manuscript; this is important

information and simply hand picking a couple of ORs to report introduces bias.

Done. Starts at Line 323, including Table 5

3. If the p value is > 0.05 (as it is with patients with dementia), please simply state there was no difference. Stating there is a “weaker signal” is misleading.

Done. Line 361

4. The descriptive information LOS, final destination of discharge, etc. is important; however, more robust analyses to determine association between patient characteristics and these outcomes would add notably more value to this work.

Done. As above

5. Much of the tables/figures are reporting of descriptive statistics (e.g., 72% of patients with rheumatological disorders underwent surgery); however, without understanding the relationship via a regression model of this factor with others, this doesn't really add much insight or help align limited resources.

Done. As above

#DISCUSSION

1. I believe the Discussion would benefit from a first paragraph that restates - in one sentence or two - the background for the study, followed by stating the goal(s) of the study, and then ending with the key findings that answer the study questions. As it currently written, a lot of good information is present in the first paragraph, but the authors already begin to compare their findings to other findings in the literature.

Edited. Line 353

2. Overall, the Discussion has a great deal of important information; however, as is, I find the Discussion to be a bit wordy and not have a logical flow. The authors discuss common areas of periprosthetic fracture, etc. but their study did not look at this specifically. It may be possible to comment on it as a reason for a result they found, but describing this in detail, to this reviewer, does not seem necessary. In reality, I believe the authors would benefit from structuring their Discussion to have one paragraph for each question/study goal stated. This would make it clearer and more concise for readers.

Edited and reworded throughout the discussion

3. Throughout the Discussion, the authors should refer to the literature; I do not think having a separate section named “Comparison with previous studies” is warranted.

Edited and reworded as suggested.

4. In addition to some of the limitations noted, I believe there are a few others that are important to consider: 1) the lack of PROMs/functional outcomes/QoL measures limit the complete understanding of the impact of periprosthetic fractures on patients/outcomes; 2) long-term results (outside of mortality linked through the national death registry) are unavailable from my understanding of the database [please correct me if I'm wrong]; 3) I agree that the lack of ability to discern between the location of the periprosthetic fracture is a large limitation; however, hip and knee are most common/likely. Can the authors cite literature noting this to be the case, so it is clear to the readers what the majority of periprosthetic fractures considered in this study are likely associated with?

Edited. Line 386. Explanation added at line 376 that it is not possible to determine anatomical location from ICD-10 code M96.6

5. The authors conclude their Discussion with good insight into potential policy implications. As noted earlier, I believe more actionable insight can be gained from this large dataset about who may or may not need greater resources, etc. This insight can help in the planning and care process more so than simply saying there is an increasing number of periprosthetic fractures.

6. Please state areas for future work/research.
Made more explicit in the conclusion, including line 602.

Overall, I believe this manuscript includes some interesting national (UK) findings related to periprosthetic fractures. However, a number of major revisions/edits, in this reviewer's opinion, are needed before it would be suitable for publication in BMJ Open. Thank you for the opportunity to review your work, and hope you and yours are staying safe during this ongoing global pandemic.

Reviewer: 3

Reviewer Name

Can Doruk Basa

Institution and Country

Tepecik Training and Research Hospital, Department of Orthopaedics and Traumatology, Izmir, TURKEY

Please state any competing interests or state 'None declared':

Sports traumatology

Knee

Shoulder

Arthroscopy

High Tibial Osteotomy

Please leave your comments for the authors below

This study haven't designed properly. Because the upper and lower extremity peri-prosthetic fractures are not similar in nature. And also, the code "M96.6" includes fractures following bone plate too. It is not appropriate to evaluate trauma patients and arthroplasty patients in the same study. Nevertheless, the title, discussion and introduction of the study flow through peri-prosthetic

fractures. Since these patients are demographic and socioeconomically different, it is not appropriate to determine their characteristics in the same study. The results of the study do not contribute to either the orthopedist or the epidemiologist from this perspective.

We would disagree that it is not possible to look at these cohorts of patients together. The code M96.6 was specifically introduced to cover all peri-prosthetic fractures (both arthroplasty prosthesis and fixation devices). Whilst the initial reason for implantation may vary, nearly all periprosthetic fractures are caused by a traumatic event such as a fall. HES data does not require specification of the limb injured so it is not possible to give definitive numbers of upper v lower limb.